# Sustainability in Public Pension Funds? A Longitudinal Study of the Council on Ethics of the Swedish AP Funds

**Joel Boudin [1] and Jan Olsson [2],*** 

[1] Kumla Kommun/Stadshus, 692 30 Kumla, Sweden; joel.boudin@kumla.se
[2] School of Humanities, Education and Social Sciences, Örebro University, SE-701 82 Örebro, Sweden
* Correspondence: jan.olsson@oru.se; Tel.: +46-708-556522

**Abstract:** Are public pension funds taking sustainability values into serious consideration? This question is addressed by analyzing annual reports of The Council on Ethics in the Swedish public pension system, which has a clear mission from The Swedish Government to consider sustainability values. The council was established in 2007 and supports four funds with advice. This article studies empirically how the council's expression of words connected to different values has changed over time as well as how it practically reasons in situations of value conflicts. The quantitative data shows that words indicative of "sustainability values" have considerably increased. As a contrast, the critical discourse analysis shows that the council often reasons in a general, loose way about preferable solutions, while more practical claims for action are largely lacking or are vague in relation to sustainable development. The underlying rationale is very much in line with the discourse of economic rationalism. Thus, the quantitative findings suggest an emerging sustainability discourse, while the qualitative analysis clearly indicates that an economic rationale continues to underpin the council's practical reasoning. However, it is concluded that this is not a simple case of green washing documents but rather a slow train moving towards green institutional change.

**Keywords:** sustainability; public pension funds; Council on Ethics; critical discourse analysis; economic rationalism

## 1. Introduction

Large financial institutions—such as pension funds—play a key role in global capitalism by financing different types of economic activities around the world. By influencing norms and behaviors in the financial market they could alter the economic rationales and behaviors of large multinational companies. For green transformative change to happen, investors need to systematically incorporate sustainability criteria into their practices and develop new norms in service of biosphere stewardship [1,2]. Public pension funds have grown over the years and provide new possibilities for humanizing the economy and make it more sustainable [3]. They are thus in the position to affect the norms and the practical rationale of the financial market and seem increasingly prone to do so in relation to sustainable development goals [4–11]. However, there is still limited understanding of the significance and potential of these changes. Are public pension funds really taking sustainability values into serious consideration? A basic hypothesis is that the value conflicts between economic growth and social and environmental values have become increasingly apparent over the last decades and triggered a more conscious handling, also in the financial system.

In reviewing previous empirical studies in the area of public pension funds and ethical investment, it can be concluded that many of them have adopted comparative case-study designs [4–9], while a few exclusively focus on one national setting [5,10]. Furthermore, most of the examined studies employed various kinds of qualitative methods with some exceptions [11]. The findings also point to a widespread recognition of ethics in public pension funds and their investment strategies. An identified "gap" in the ethical investment

literature is the lack of systematic longitudinal examinations, focusing on how public pension funds change over time in terms of policy language and underlying discursive framings. In relation to this, none of the reviewed studies employed a mixed method design in the sense that quantitative findings are compared with qualitative results, thereby making it possible to illuminate discrepancies between developments in the manifest and latent level of documents.

Moreover, knowledge appears to be scarce in connection to the bridge between traditional public and business-oriented ethics, which is particularly interesting when pension funds are public organizations and operate on a market. The point of conducting a study that intentionally targets this "bridge" between public, environmental and business ethics is that value conflicts can be assumed to arise, forcing the organization to make choices and to advocate certain solutions at the expense of others. This can further tell us something about the actual ethics and values of that organization; that is, how it acts when facing a pressing situation in which multiple interests and values are at stake.

Against this background, the Swedish public pension system is a good case in point. In Sweden, a new organizational unit was established in 2007 called The Council on Ethics, which was created for the purpose of coordinating the ethical and environmental work of the AP Fund 1–4. The four competing funds received the same mission in 2001—to place the pension capital in order to achieve the greatest possible benefit for the pension system and to create high returns at a low risk in the long term. The funds ought to consider ethics and environment, without renouncing from the overall goal of high returns. As of 1 January 2019, a new set of rules and goals was introduced for the AP Funds by the Swedish parliament, including the goal of managing the pension capital in a way that contributes to sustainable development [12,13].

In the aftermath of this goal being introduced, an intense media debate took place in which the AP Funds were described in terms of circumventing the rules that obligate the funds to integrate environmental aspects into their investments. For instance, a prominent media outlet called *Sveriges Radio* wrote that: "Since 1 January, the AP Funds have an obligation to consider the environment in their placements. However, critics suggest that the funds could circumvent the environmental demand due to the funds principal task of gaining financial returns" [14] (p. 1). The newspaper *Expressen* highlighted a report produced by *The Swedish Society for Nature Conservation* (SSNC) (*Naturskyddsföreningen)*: "Children suffer brain damages, native populations are being moved by force, and entire shantytowns sicken. Despite brutal testimonies in the shadow of the fossil giants the companies are being financed with billions from Swedish pension beneficiaries" [15] (p. 1).

Considering the vital role of public pension funds and the research gaps regarding how institutional values change over time, the focus of this article is to provide a longitudinal empirical study of the language use of the Council on Ethics and how it makes sense of and reasons with respect to value conflicts in their investment activities.

The aim of this article is to examine the ethics of The Council on Ethics by empirically studying how its expression of words connected to public, environmental and business-related values have changed over time as well as the practical reasoning of the council in situations where value conflicts are present. What actions does the council advocate for in terms of practical solutions to these conflicts? Moreover, the analysis seeks to detect the discursive framings of the practical conflict reasoning. This unfolds into the following research questions:

(1) How has the expression of words connected to public, environmental, and business values changed over time in the annual reports of the Council on Ethics?
(2) How does the council reason in the context of value conflicts? What are the practical arguments in favor of certain actions or solutions and what are the underlying discursive rationale of this argumentation?
(3) In relation to questions (1) and (2), are there any discrepancies between the developments in the usage of words and the practical reasoning of how to understand and handle value conflicts? If yes, in what sense?

The empirical results in relation to these questions will be discussed and elaborated on in the concluding section, as will some normative implications and the need for future studies of greening public investments.

## 2. Political Discourse and Practical Reasoning

This article combines three theoretical perspectives within the broad discourse theoretical tradition. The overarching perspective/method is developed by Norman Fairclough and Isabela Fairclough [16] and is called "practical reasoning". Two highly relevant discursive perspectives for the theme of this article—the sustainable development discourse [17] and the economic rationalism discourse [18]—will help us generate two hypotheses that will guide the empirical analysis.

### 2.1. Discourse and Critical Discourse Analysis

In the literature, discourse is commonly understood as a particular way of framing reality by language through the construction of concepts and perceptions of what reality looks like [19,20]. For this study, the analysis of *political discourse* will be in focus, which according to Fairclough and Fairclough can be regarded primarily as an "argumentative discourse", as well as being "based on a view of politics in which the concepts of deliberation and decision-making in contexts of uncertainty, risk and persistent disagreement are central" [16] (p. 17). Even though this framework was developed for and has been mainly used for traditional political discourses, like parliamentary debates, it is also suitable for other discourses pervaded by value conflicts, such as public deliberation over complicated investment decisions [16].

According to Norman Fairclough, the critical element of discourse analysis alludes to normative critique of "social wrongs", which in turn takes its departure from a certain set of normative values. One of the main goals of this critique is to assess the potential gap between what "societies claim to be ('fair', 'democratic', 'caring' etc.) and what they are" [21] (p. 7). Another kind of critique is explanatory, which refers to explaining why some "social orders" are upheld through discourse over time, despite being harmful or unjust in nature. Explanatory critique can also allude to explanations of changes in social order; which causes and mechanisms that enable this change [16] (p. 79).

For Fairclough and Fairclough [16] (pp. 17–18) practical reasoning is a central concept in their version of political discourse analysis, which focuses on *action* in various political contexts. Practical reasoning connects to the issue of "what to do" in given situations under certain circumstances in achieving different goals. The issue of how to act in a situation can be informed by the goals and values of an agent, but also by the institutional and social context. In connection to the aspect of institutional context, Fairclough and Fairclough argue that one should separate between *actual concerns* and *socially recognized* value commitments or moral norms. Actual concerns or values motivate the agent to realize his or her goals, while socially recognized norms and values do not have to lead up to action [16] (p. 48) (see Figure 1). Hence, for socially recognized norms and values to be turned into action, these need to be "internalized" by the agent in the sense that they become "real psychological motives" [16] (p. 42).

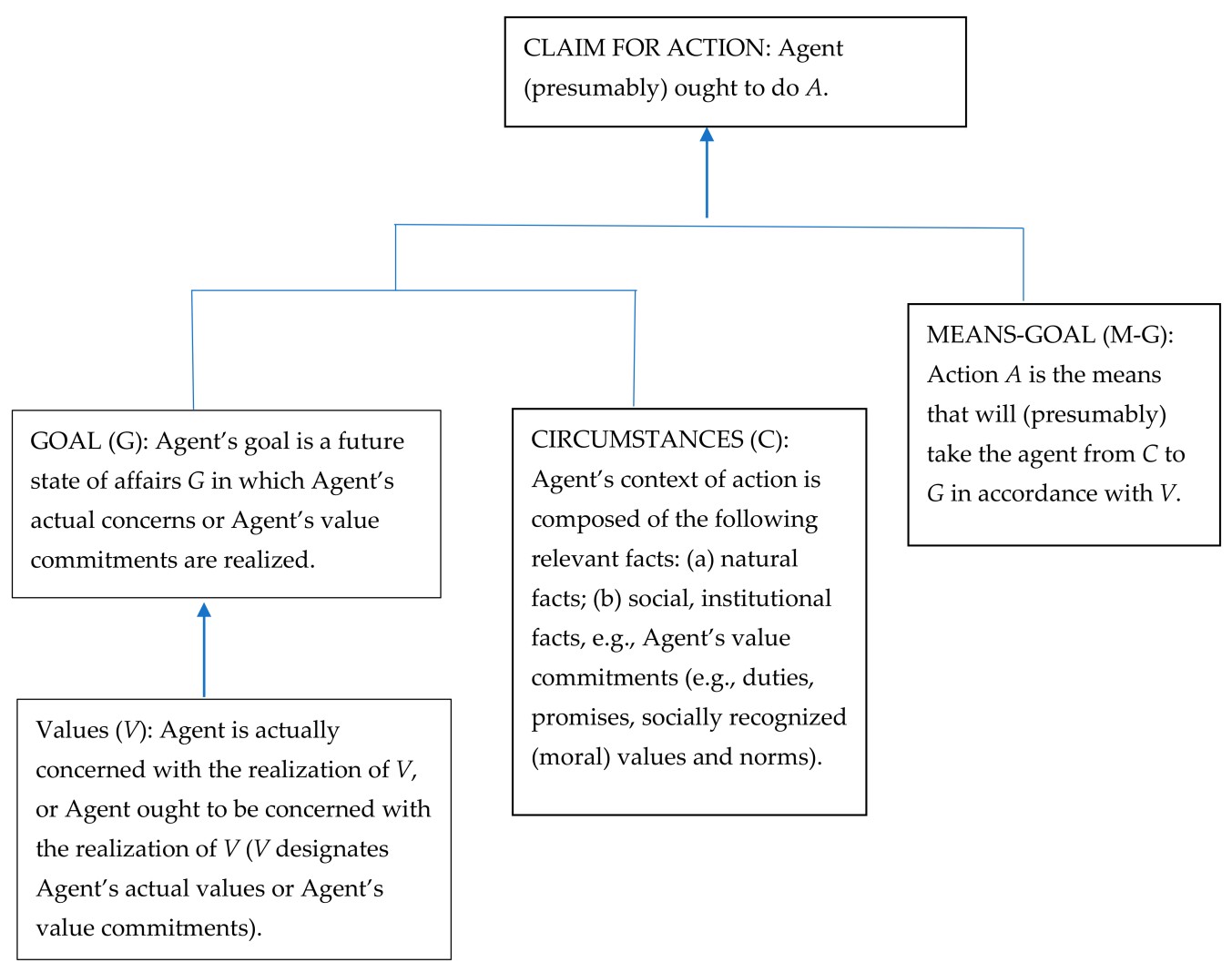

**Figure 1.** An illustration of practical reasoning, from Fairclough and Fairclough [16] (p. 48).

In practical reasoning, the goal of an agent is considered the "major premise", meaning that this constitutes the core of the proposed action. Another part of practical reasoning is the circumstantial premise; that is, the institutional and social context with its associated norms and value commitments [16] (p. 44). When an agent is facing a situation that presents alternative ways of acting, and neither of these alternatives appear to be inherently wrong, a process of critical testing is undertaken in which various actions and their respective consequences in relation to prioritized goals are imagined on a theoretical level. In this way, practical reasoning includes a deliberative dimension in terms of weighing arguments, goals, and values against each other to decide which of the solutions that is the most reasonable one [16] (pp. 47–50).

The concept and framework of "practical reasoning" not only concerns a descriptive approach to making sense of arguments, but also includes elements of critical discourse analysis [16] (p. 85). One such element is to analyze "representations of reality", which alludes to various ways of describing the world and making sense of it. Furthermore, since individuals, and organizations make sense of the world in different ways and argue for their own perceptions to convince others, representations of reality should be critically examined as "parts of premises" in practical argumentation. In this way, the analysis of representations connects to the understanding of how and why agents advocate certain actions [16] (pp. 92–94).

The framework of practical reasoning described above will be used to analyze the practical argumentation of the council in terms of what ought to be done in situations

where conflicting values are present. In this way, we expand the theoretical model of practical reasoning in relation to the component of "circumstances", where value conflicts are added as a basic contextual factor. Hence, the analysis will attempt to highlight the proposed actions of the council in the context of value conflicts, but also to examine how these proposed actions are connected to certain discursive framings or representations of reality. Considering the overarching mission of the AP Funds, one could expect economic rationalism—in terms of market analyses, calculation, and profit-seeking—to pervade the practical reasoning of the council. However, considering that sustainable development is an increasingly important discourse in society, and that the Swedish Government's AP Fund policy explicitly applies this discourse, it is likely that the council's practical reasoning reflects this by openly discussing and balancing conflicting values, in particular those of great relevance to sustainable development. Thus, this article will use the practical reasoning approach of Fairclough and Fairclough and apply two discourses that are expected to be important in the practical reasoning of the council.

### 2.1.1. The Discourse of Sustainable Development

Sustainable development is a widely spread ethical discourse today. Ethics are commonly perceived as "values and principles that guide right and wrong behavior" [22] (p. 9). Values are ideas or things that people attach value to, while principles guide actions in various situations. Thus, ethics is not just about ideas of what is right and wrong, it is also about concrete actions and consequences. When a set of values and principles are established and implemented within a public organization, they jointly form the ethics of that organization.

Sustainable development is a broad discourse with internal tensions. It has been depicted as an essentially contested concept, in the meaning of William Gallie [17,23]. The concept is multidimensional (ecological, social, economic) and consists of four key principles: intergenerational justice, global justice, ecological justice, and participation [24]. There is also a tension between these principles and how they are applied and implemented in practice [25]. The term sustainable development is regularly used by both "radical greens" and "capitalists", and the former group accuses the latter to use it to conceal conflicting interests between environmental values and economic growth [17] (p. 22). Another point of conflict is captured by the distinction between instrumental and intrinsic values, implying that the nature can be valued either as a resource for the benevolence and well-being of humans or that the nature has intrinsic values decoupled from human needs and desires ([26,27], (p. 12, p. 116)).

In a more concrete sense, sustainable development as a discourse is about discussing, making sense of, and managing different values in specific situations and seeking to find a balance between contradictory values or to prioritize one value before the other with help of explicit arguments. Sustainable development in relation to the AP Funds needs to be understood both in relation to the value-complexity that the funds are facing and the actor-complexity in terms of goals and instructions from the Swedish Government, including signed international conventions; international organizations like the UN; business organizations, and a large number of private companies. All these actors have potential influence over the views and practical reasonings of the funds and their council. From the sustainable development discourse, the following hypothesis is generated:

**Hypothesis 1 (H1):** *In the context of value conflicts, the council will make practical arguments for action that explicitly consider and deliberate on sustainable development values, leading to well-motivated investment judgements/decisions.*

### 2.1.2. The Discourse of Economic Rationalism

In the words of John S. Dryzek, "Economic rationalism may be defined by its commitment to the intelligent deployment of market mechanisms to achieve public ends" [18] (p. 122). This can be applied to serve various environmental ends, for instance by using

charges and taxes to induce decreased emissions and congestion. The underlying reason for using market-related instruments to solve environmental problems is that it is supposed to entail smaller costs and to offer effective solutions.

Dryzek discerns four interrelated components in economic rationalism. First, it assumes basic entities whose existence is recognized or constructed, such as the market. Human beings "appear as a consumer or producer; and if producers are organized into firms, the firm still behaves like an individual. Markets, prices, and property have real existence" [18] (p. 134). Nature, along with ecosystems and natural resources, is constructed as an instrument or pathway for human actions and decisions, thereby serving the needs of social and economic systems [18] (pp. 134–135). The second component of economic rationalism assumes anthropocentric relationships between humans and the environment: "nature exists only to provide inputs to the socioeconomic machine, to satisfy human wants and needs" (2013:135). This view also consists of the idea of a state of competitiveness between human beings and organizations, and that nature is subordinated to this competition [18] (p. 135). Natural resources are thought of as private properties that ought to be traded and sold. This idea is further underpinned by the perception that natural resources are "finite", which implies that there are incentives to create a market system [18] (p. 135). Third, in economic rationalism individuals and organizations are perceived as motivated by material self-interest and pursuing it rationally [18] (p. 136). Hence, the world is principally populated by consumers, producers, and sellers, not by citizens and animals.

The last component of economic rationalism is key metaphors and other rhetorical devices, which help frame and portray certain actions and events in society as harmful or beneficial, depending on their implications for the market. One rhetorical framing that is recurrent among economic rationalists is "command as control", which is connected to governmental interventions and the associated perception that public regulations are bad and ineffective [18] (p. 136). Another rhetorical device concerns the emphasis on freedom in relation to the market; the market is "free" and thereby becomes automatically desirable [18] (pp. 137–138). From this discourse we generate the following hypothesis:

**Hypothesis 2 (H2):** *In the context of value conflicts, the council makes practical arguments for action that are in line with the economic rationalism discourse, both in terms of practical solutions and their motivations.*

## 3. Materials and Methods

### 3.1. Case Selection: The Council on Ethics

There are six AP Funds in the Swedish pension system. The Council on Ethics was set up in cooperation between the AP Funds 1–4. These four funds were in 2019 responsible for 160 billion Euro. The council works as a coordinating body of the four funds in connection to their ethical and environmental considerations. The task of the council is to influence the portfolio companies of the AP Funds to integrate and implement aspects of sustainability, for the purpose of making sure that these companies conduct their businesses in a responsible manner. The council also gives advice to the four funds about investments and disinvestments. Within the council, one permanent member from each of the AP funds is represented. Thus, the council and the funds are closely connected, which implies that the council advice is well anchored among the funds. Furthermore, since 2007, when the council was formed, annual reports have been produced in which the activities and objectives of the council along with the AP Funds are accounted for. These annual reports also constitute the empirical data for this study, allowing longitudinal study. The AP Funds are public agencies, and their documents are public documents under the Swedish principle of openness [12,28].

The council can be perceived as a critical case that may allow analytical generalizations, that is, if a theory works in the conditions of the critical case, it is likely to work anywhere, and vice versa: "If it is valid for this case, it is valid for all (or many) cases", and conversely

"If it is not valid for this case, then it is not valid for any (or only few) cases" [29] (p. 230). Considering that Sweden is often portrayed as one of the frontrunners when it comes to sustainable development [30,31], it could be expected to be true also for the council. However, if it is impossible to validate the prevalence of the sustainable development discourse in the Swedish pension funds and their council, then it is probably not valid for any other case either.

The annual reports of the council that will be examined in this study are the ones from 2007, 2012 and 2018, which makes it possible to investigate longitudinal developments. Moreover, this selection is grounded in the fact that the combination of quantitative and qualitative studies is quite demanding, which makes it is necessary to delimit the empirical data to make it possible to analyze it in a systematic and thorough manner. The excerpts used for analysis have been translated by the authors from Swedish to English.

### 3.2. The Quantitative Analysis

In studying how the usage of certain words has changed over time a quantitative content analysis is employed. This is systematic analysis of textual information, which is done by applying a set of predefined categories onto the data in a consistent manner so that the risk of bias is reduced as much as possible [32] (p. 345). Reducing bias is also connected to transparency, to show how the categories have been applied and that this process is in accordance with the established rules created in advance [20] (p. 289). Furthermore, quantitative content analysis is chiefly concerned with the manifest level of texts, examining the frequency of explicit expressions.

In creating rules for the analysis, and in this way establishing reliability and consistency, a coding manual was made. This manual explains how each category is to be applied in relation to the data, which in this study is connected to certain words that indicate the presence of categories. The categories with belonging words have further been labeled with codes, which can be described as shortened tags that represent the categories. To exemplify, the category *transparency* is coded as (T) and the second word under this category is "open"; hence, the code T2 ([19,20,32], (p. 55, p. 299, p. 349)).

Since words can be written with different inflections, each word in the coding manual is marked with a star (*), meaning that the same word with another inflection is included in the analysis. Moreover, words of a more general nature were included, such as "environment" and "social", since these can provide some indication of how values have changed over time. When measuring these general words, they can also constitute a part of phrases of a more general nature, for instance "social aspects". Hence, each general word is followed with three dots in the coding manual to indicate that these words can be followed by something more.

To find the relevant words in the documents, the navigating function in the Word software was used. This function enables a quick search for a specific word and immediately shows the total number of times that this word is mentioned in the document. It also shows the sentences in which this specific word is mentioned, which puts the word into context. This made it possible to determine whether words of a general nature were relevant or not in relation to the various categories. Besides using Word, software called Wordsmith was used to create wordlists of the studied documents. These wordlists contain all words in the documents and show the frequency of each word. Furthermore, to be able to make comparisons between the documents in terms of how the usage and frequency of certain words have developed over time, the total number of words for each variable was calculated against the total amount of words in each document, thereby coming up with numbers in percentage for the variables.

### Keyword Analysis

To get an accurate idea of the characteristic words of the studied documents, a keyword analysis was made with Wordsmith. Wordsmith can be described as a "corpus-linguistic software" used to generate so called "keywords" [33] (p. 12), which is a common method

in social science. (For further reading about Wordsmith and Keyword analysis: [34,35]). In this case, this means that one large wordlist was created based on all the annual reports between 2007–2018, except for three reports (2008, 2014 and 2016), which were not possible to convert into the right format for Wordsmith. Then, each of the chosen reports was analyzed for keywords in relation to the large wordlist. In this way, it was possible to calculate which words were mentioned at an unusually high rate in one document, using the remaining documents as a point of reference. The documents analyzed for key words were the reports from 2007, 2012 and 2018, selected to maximize the longitudinal character of the study.

*3.3. The Critical Discourse Analysis*

The critical discourse analysis is a qualitative study of the practical reasoning of the council in situations of value conflicts. It follows the approach and method of Fairclough and Fairclough [16], presented in Section 2.1. The selection of value conflicts within the annual reports (2007, 2012 and 2018) follows a logic similar with the notion of "critical cases". This refers to a strategy of selecting value conflicts between business values and public or environmental values, since these values conflicts are "most likely" to reveal if the practical reasoning of the council is more clearly underpinned by the discourse of sustainable development or economic rationalism [16] (p. 231). The selected conflicts are analyzed in relation to the two hypotheses to give both fair chances to get support. Furthermore, the critical discourse analysis will illuminate alternative ways of perceiving or portraying the problems and logics embedded in the reasoning of the council. The analysis will include both explicit reasoning and implicit such, since the "naturalized implicit propositions" can constitute an important element of discursive framings [21] (p. 26). To make the analysis transparent, relatively long citations from the annual reports are used.

**4. Results**

*4.1. The Quantitative Analysis*

The quantitative content analysis—which is visually presented in Figure 2 below— shows that the amount of words indicative of financial business values has decreased, which is also the case for public procedural values. In terms of magnitude and significance, this development stands out. Moreover, it can be deemed interesting that words connected to financial business values and public procedural values have decreased more significantly than other variables, especially considering their weak theoretical relationship. Another development that should be recognized is that words related to sustainability values have increased quite dramatically over time. What this increase represents can be discussed. One interpretation is that this is a broader discursive change in which words connected to the concept of sustainability are used as a way of gaining legitimacy, particularly when the organization is subject to "new" sustainability demands from their superiors. However, it can also be interpreted as an unintentional increase in the usage of such words. The concept of sustainability has been popularized over the years. It may thus have the psychological effect of being continuously integrated into the minds of both public and private actors, which in turn affects the use of language. However, it should be emphasized that the exact extent of what is intentional and unintentional is difficult to determine, but it can be fruitful to think in such terms, since discourses tend to be both "constitutive" and socially or externally constituted [19] (pp. 356–357) (see Figure 2).

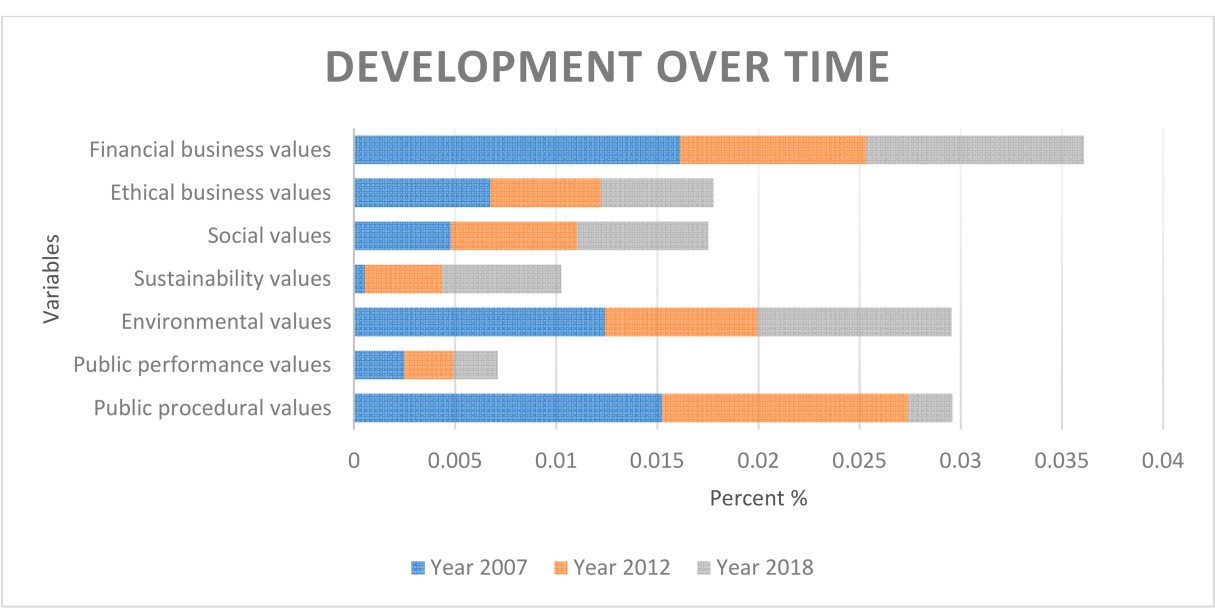

**Figure 2.** Diagram displaying the percentual developments over time in terms of words indicative of values connected to public ethics, environmental ethics, sustainable development, and business ethics.

In the keyword analysis, it can be recognized that ethically oriented terms such as "sustainability", "climate" and "labor rights" are characteristic of the 2018 report, which is not the case for the reports from 2007 and 2012 (see Figures 3–5).

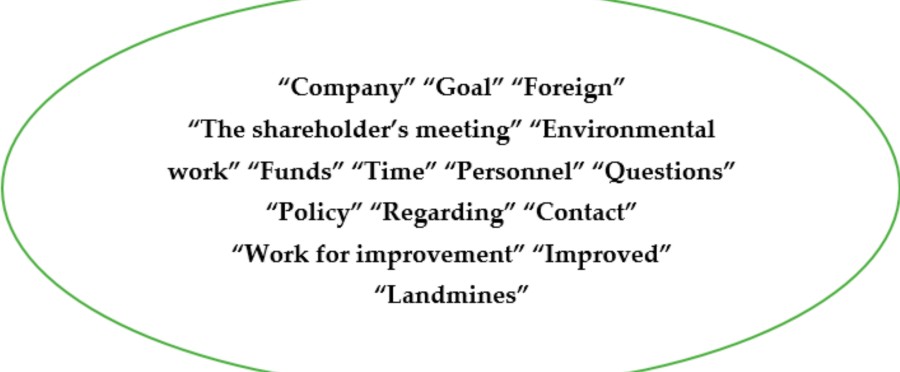

**Figure 3.** Keywords of the 2007 Annual Report.

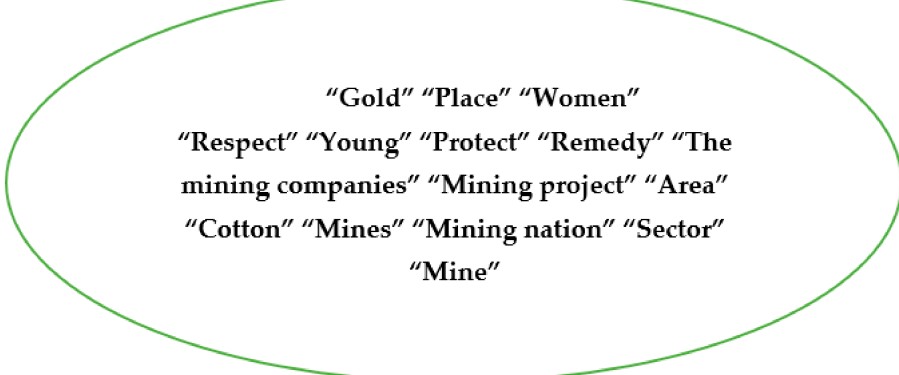

**Figure 4.** Keywords of the 2012 Annual Report.

"Company" "Global" "Sustainability goals"
"Security" "Climate" "Deforestation"
"Sustainability" "Proactive" "Narcotics" "Health"
"Living salary" "Business ethics" "Environment"
"Labor rights" "Research"

**Figure 5.** Keywords of the 2018 Annual Report.

One interpretation of this result is that the council has become more "ethical" in its language use over time, since "sustainability", but also to some extent "climate" and "labor rights", can be viewed as concepts rather than plain words. They represent something more than "ordinary" terms whose only function is to describe or illustrate. In line with this reasoning, words such as sustainability are followed by more abstract ideas of what sustainability is and how it can be achieved, which in turn can be connected to the socially conditioned production and reproduction of discourse. The point being made is that the above-mentioned words are not likely to be systematically used by the council for no reason, they come with ideas and perceptions adhering to more fundamental views of reality, views that can originate from discourses positioned outside of the immediate context or views that are embedded within the language of these documents.

*4.2. Critical Discourse Analysis: Practical Reasoning of the Council on Ethics*
4.2.1. Sustainable Development Ethics?

How does the council reason about conflicts in relation to sustainability values? In all three annual reports (2007, 2012 and 2018), sustainable development is addressed, but in rather general terms. The council does not elaborate on the concept in relation to the degree of radicalism or in terms of intrinsic values. In this respect, the reasoning is similar in all the reports. There are also two other, largely contradictory patterns in their reasoning: principle compliance and local solutions. These patterns are presented and analyzed below.

In the beginning of the annual report from 2007, the chairman of the council describes her view of the long-term goals of the council's work:

"If gazing towards the future I see continuous development for the work of the Council on Ethics, and increased cooperation. All the funds support the UN-initiative Principles for responsible investments (PRI), and there we see increased activity, engagement, and opportunities for investors worldwide to cooperate for a sustainable development. The government appointed a committee in 2007 that will investigate the guidelines of the AP Funds in terms of environment and ethics. With this report, we want to show all our stakeholders that environment and ethics are on the very top of our agenda" [36] (p. 1)

The quote above can be interpreted as a "claim for action", since the act of writing the report can be viewed as a solution to the "problem" that the AP Funds are subjected to a governmental investigation. As pointed out above, the mentioned values can be considered externally imposed onto the council and the AP Funds, which according to the theory of Fairclough and Fairclough [16] (p. 48) can entail a motivational problem. Moreover, the act of showing stakeholders that the council incorporates environmental aspects and ethics can be interpreted as means to achieve recognition as responsible investors.

Going further, the council describes "the dialogue as our most important tool to make companies act responsibly" [36] (p. 4). The council explains that:

"The Council on Ethics wants to spread knowledge and create an understanding for the work that we conduct. Therefore, we have chosen to go out with the names of the companies that we have a dialogue with today, and what goals we wish to accomplish in relation to each individual case. However, we have chosen not to account for any details from on-going dialogues since their success depends on the existence of trust between the Council on Ethics and the companies we are discussing with." [36] (p. 5)

In this reasoning, one can identify a deliberative element in the sense that the council faces two conflicting values: public transparency and company trust. On the one hand, the council must show the government and other stakeholders that they incorporate environmental and other values in their work; on the other hand, they want to maintain integrity in the dialogues with portfolio companies. Consequently, to balance these conflicting values, the council ends up in a compromise in which transparency and accountability are fulfilled to some extent.

In the annual report from 2012, the council provides an account of their so called "mining project" that relates to "issues of sustainability" among mining companies and the conducted dialogues aimed at improvements within this area [37] (p. 8).

"The global demand on gold is driving the new prospecting as well as the fact that many historically significant deposits of many minerals have declining production. This creates a situation in which mines are prospected for all over the world, mines that create working opportunities and well-needed national revenues but also competes with local traditional lifestyles and create concerns of pollution and access to water and other environmental impact. To gain and keep the trust of the local residents is very important. In turn, the companies are in the dialogues handing the general impression that they are aware of this, but they declare different methods and approaches to handle the situations, based on the companies often local experiences. This is probably a wise strategy since every mine has its own unique social and environmental challenges but the need for exchange in experiences between companies is plausibly healthy and something that the Council on Ethics encourages" [37] (p. 9)

In the passage above, the claim for action is that mining companies should adapt their methods and approaches according to the local circumstances and experiences, suggested as "probably" being a "wise strategy". This can further be connected to the goal of creating harmony between possibilities of economic prosperity and environmental/social risks, based on the value commitment of trust among locals. If lingering on the expressed value of creating trust among local citizens, it remains implicit how this trust ought to be realized. Hypothetically, mining companies can promise economic prosperity and revenues for the local residents and thereby creating short-term trust, but if the long-term consequences of mining entails severe pollution and less access to water, this construction of trust can be questioned in terms of principally favoring the mining companies. Accordingly, in this context, when important values are at stake, trust can be interpreted as a relatively "weak" value in the sense that access to water and a healthy environment are fundamental aspects of human life that cannot be subjected to compromise or neglection.

Furthermore, the circumstantial premise (value conflict) in this reasoning can be interpreted as being that mines are sources of prosperity and revenue but also entail social and environmental hazards, as well as each mine being unique in terms of having its own "*challenges*". Therefore, it is assumed that the companies ought to find their own solutions based on local conditions. This reasoning can be deemed as a contrast towards the generally strong emphasis on conventions and compliance with international rules and principles from the UN, which is indicated by the quantitative study but is also is formulated by the council itself:

"Together with the basic values of the Swedish government, the international conventions that Sweden has signed and Sweden's positions in issues of public

international law constitute essential instruments for the council in its work" [37] (p. 6)

Accordingly, this logic can be interpreted as contradictory in the sense that companies should develop methods and tools in accordance with institutionally recognized principles and guidelines, but also find their own solutions based on local experiences and circumstances. Moreover, if addressing the selection of words in the passage above, the terms "probably" and "plausibly" do not contribute with any clarity regarding how companies should relate to the divide between local and international responsibilities. This contradictory reasoning, along with the selection of vague terms, can confuse companies and stakeholders with respect to what the point of reference actually is for the council, as well as creating unclear situations in which local values risk to be put up against international UN principles. To illustrate: if imagining a scenario in which a local solution was found by a mining company that was good for local residents in terms of economic prosperity but violated various principles of environmental protection, how would the reasoning of the council contribute in solving this?

In connection to the above-mentioned mining project, the council accounts for their visit to Burkina Faso which is described as "one of the poorest nations in the world" [37] (p. 10). More specifically, the visit concerned the country's mining industry, reporting that this industry creates "revenues to the state and working opportunities", as well as claiming that "it is working opportunities and revenues that are necessary to begin the journey out of poverty" [37] (p. 11). At the same time, the council points out that the mining industry faces challenges in terms of environmental and social issues:

"Another important question that the companies must manage responsibly is the social and environmental risks that arises. The impact of the mining industry on environment and society is extensive, and here the mining companies have a big responsibility"

" . . . to have a good dialogue with the affected local residents and consulting them both before and during the process is an important factor for success", but also that "Burkina Faso has a big number of different ethnic groups with partially different needs. Simply, there is no panacea that suits or appeal to everyone" [37] (p. 11)

Again, the council makes a claim for action based on local experiences and solutions. Companies are encouraged to have "good dialogues" with affected citizens and to adapt dialogues to ethnic groups. Viewed in isolation, this claim can be considered reasonable and constructive, but if looking at the annual report in its whole and the relatively high number of legally related terms (LA = 74), as well as looking at the council's advocacy and use of UN guidelines and principles as basic instruments, a tension can be identified between the argument for local solutions and internationally recognized approaches based on global principles. Moreover, with respect to the claim that mining companies must manage the social and environmental risks, one can wonder whether these risks ought to be exclusively managed within the local arena based on local experiences, and to what extent external or international interests can influence this risk management.

Neither is the underlying value conflict explicit in the reasoning; the mining industry is described in terms of being a source of employment and economic prosperity for the local residents as well as other parts of the population, without being depicted as a possible impediment in relation to the management of negative environmental and social risks. Here, one can argue that the there is a value conflict that needs to be more outspoken and explicitly addressed, namely that the mining is a source of economic prosperity, but also entails ethical and environmental risks that may be neglected to some extent. The logic of the reasoning is more oriented towards putting the trust in companies taking their responsibility through risk management.

To summarize, this analysis shows, in line with the quantitative data, that the council explicitly addresses sustainable development values, but its practical reasoning does not

seem to be driven by an ethical position of their own, developed through internalization of sustainability values. Instead, the council's main strategy is obviously to anchor its judgments by motivating and adapting in relation to external actors and rules, such as the Swedish Government and the UN. This top-down reasoning is, however, complemented by practical reasoning on how and why local contexts should matter, which seems contradictory and tends to relativize the council's main position on sustainable development.

### 4.2.2. Economic Rationalism?

Considering that the overarching goal of the pension system is to generate economic return to the pension beneficiaries it does not come as a surprise that the economic rationalism discourse is of central concern and is also persistent in the annual reports over time. In the following analysis, this consistent nature of reasoning will be illustrated by several examples.

In the annual report from 2007, the council provides an account of their visit to China in which ethical and environmental standards of portfolio companies are in focus. The council explains that China has had "huge environmental problems" during recent years and that fossil fuels constitute one of the key aspects of this problem:

> "As the situation is today, around 80 percent of China's energy comes from fossil fuels (primarily coal) which leads to environmental pollution both in relation to coal mining (heavy metals) and big amounts of carbon dioxide emissions. According to the World Bank, the pollution of air and water costs 5.8 percent of China's BNP, above all because of the extensive cases of diseases and deaths" [36] (p. 12)

In the long term, change must be carried through in terms of finding alternative sustainable sources of energy. However, the council's rationale for change is not only connected to various harmful environmental impacts, but also to an economic logic that appears to treat the environmental and social consequences as a path towards financial instability on the national level, explicitly mentioning the share of China's BNP that goes lost in paying for the social effects of continuous pollution. Hence, the intrinsic reason to act for a change in the environmental area is related to the economy. This can be seen in the light of Dryzek's component called assumptions about natural relationships, which treats nature as subordinate to the economic system. Furthermore, in terms of practical claims for action to manage the above-mentioned value conflict, the council does not provide any clear propositions other than:

> "After the visit to China, it feels very motivating to continue to work for better environmental and working conditions in the country. It is obvious that foreign clients and investors play a big role in advancing the Chinese development within these areas" [36] (p. 13)

Here, it becomes clear that the council has not come to any concrete conclusions in terms of what actions companies and investors should implement. Instead, the conclusion is formulated in a broad evaluative sense, that there are problems to work on and that the situation needs to improve in general terms.

Further in the 2007 report, the council addresses the issue of transparency in the oil industry, describing that:

> "The need for transparency and governance is especially big in countries that are rich in natural resources but weak in governing. Clearer reporting of incomes by the host nations, as well as companies reporting what they pay, increases transparency in society and contributes to better conditions for financial governance." [36] (p. 15)

The contrast of oil being a natural resource of great relevance for continuous growth as well as for climate change is left implicit in the reasoning of the council. An underlying value conflict can be identified that may or may not be related to the claim for action,

namely that transparency ought to be improved. Moreover, it can be recognized that the goal of increased transparency is connected to a financial logic; companies and host countries should be more open about their revenues and payments, so that the AP Funds and other investors can engage in "financial governance". Hence, the rational of being more transparent does not appear to be connected to the goal of gaining better insights of considerations related to environmental protection and sustainability. Building on this, the council does not explain any further how the desired state of improved "conditions for financial governance" ought to play out in practical terms.

If connecting the analysis above to Dryzek's economic rationalism, aspects such as "incomes" and "revenues" are recognized as important basic entities that need to be accounted for and managed. In contrast to this, "natural resources" and other environmental aspects does not seem to be attributed any real existence in terms of being valued independently from the financial system.

In the annual report from 2018, one area addressed is palm oil and the related production and chain of delivery. The council explains that: "In many of the portfolio companies of the AP Funds, palm oil is an important raw material. In countries that produce palm oil this industry offers new sources of revenue and opportunities to enhanced standard of living" [38] (p. 27). The council further provides a lengthier elaboration of the challenges connected to palm oil production:

> "The industry is associated with big challenges in terms of how smaller agricultures should be encouraged to enhance their methods of production as well as how traceability of palm oil fruits should be accomplished through the chain of delivery. One common way of increasing the production is to increase the acreage of plantation trough deforestation by wildfire. The wildfires contribute to the global warming and to the destruction of the rain forest, which is a globally important sink of greenhouse gas. Moreover, the dressing of the land often contributes to further emissions of greenhouse gases. The deforestation entails a decrease in biodiversity" [38] (p. 27)

Moreover, the council argues that "Sustainable production requires knowledge and investments", as well as pointing out that "In order for the share of certified sustainable palm oil to increase, it is required that existing plantations are managed more efficiently but also that there is a demand and a will to pay for sustainably cultivated palm oil. Unfortunately, the demand for certified sustainable palm oil is relatively low" [38] (p. 27).

The solution for solving the unsustainable situation described above, where important values connected to the climate, forest preservation, and biodiversity are neglected in favor of economic gains and revenues, investments and financial means can be deemed as only addressing the surface of the problem. Various actors can invest large amounts of money into sustainable palm oil, but that does not necessarily solve the problem unless farmers and companies understand why they must change their methods. In connection to this, the "demand" on the market is explicitly being mentioned as a condition for the rise of sustainably produced palm oil, which can be interpreted as a discursive framing by the council in terms of making a claim of what reality looks like. This framing is formulated in a straightforward and relatively simple manner: if there is not a demand on the market, there is no room for sustainably produced palm oil.

Hence, the market as a "basic entity" constitutes a precondition for things to happen. A certain rhetorical element can also be recognized in the sense that the market is portrayed as a regulator, not completely different from the function of governments and laws. Moreover, the component of "agents and their motives" can be identified in the sense that *production* is used systematically to describe the process of planting the fruits and subsequently extracting the palm oil, which also involves farmers that make this process possible. However, in the account provided by the council, these farmers appear to be implicitly categorized under the label of "production", without their significance as human beings or citizens being seriously recognized. In this way, farmers become "agents" whose only function and purpose is to produce and contribute to the economic system.

Going further in the 2018 report, the council addresses the extraction of cobalt in the Democratic Republic of Congo, describing the situation in the following way:

"Cobalt is an important raw material that enables the transformation to a low fossil society. Cobalt exists in lithium ion batteries and is used in electric cars, mobile phones, laptops, and other home items" [38] (p. 30)

With respect to challenges with the extraction of cobalt, the council points out that:

"Cobalt is often sold via a complex chain of delivery that starts in the Democratic Republic of Congo, where more than half of the world's cobalt is extracted. Severe systematic violations of human rights including child labor is common. The demand for the mineral cobalt is expected to grow during years to come" [38] (p. 30)

"One challenge that the investors are worried about is that some companies have stopped or have thought about stopping purchases from smaller mines entirely. Such a stop in purchases could entail big negative effects on the possibility of local people to provide for themselves" [38] (p. 30)

The reasoning of the council can be interpreted as contradictory if looking at the explanation of circumstances in relation to the claim for action. To make this logic explicit: Companies should assess the risks and take action if human rights are systematically violated; however, they should not take action in terms of stopping completely with their purchases of cobalt, since this entail negative consequences for the locals in terms of livelihood. Here it becomes clear that there is a conflict between two circumstantial premises, namely violations of human rights and loss of livelihoods. It also becomes clear that the council has not formulated any desired actions that addresses this conflict other than assessing the risks and participating in various initiatives, which is vague and hollow in terms of practical meaning.

Another part of the council's reasoning that illuminates why the AP Funds are "worried" about the stop in purchases among some companies is the statement that "The demand for the mineral cobalt is expected to grow during years to come". Accordingly, if portfolio companies stop purchasing this mineral, while the demand on the market is high for cobalt, investors are likely to perceive this as a problem in strictly financial terms. Moreover, in connection to the statement that the demand for cobalt is expected to grow, the market appears as a basic entity that underpins the reasoning of the council, it is also used as a rhetorical device in the sense that the demand works as a fundamental mechanism; if there is a high demand on the market for a product, this product or material should not be decoupled from the financial system.

Regarding the practical claims for action in the council's reasoning, these could become clearer in relation to what relevant companies ought to do. In other words, the council could have presented a more explicit structure of reasoning about the above-mentioned conflict that includes different goals and values. In such a structure, which could be used by companies to facilitate the process of assessment, the reasoning could be a critical deliberation of advantages and disadvantages of acting in one way or another, and leading to a well-informed claim for action.

The council further reports about the issue of repellants and pesticide and its damaging effects on bees, which holds an important function within various ecosystems. The council describes it in this way:

"Pollination is an important ecosystem service that is necessary for the reproduction of plants. It enables around 35 percent of the global production of cultivated crops. In the US, the honeybee is used for the pollination of around 80 percent of all plantations and is calculated to contribute with 20 billion dollars to the American economy" [38] (p. 24)

Based on the account above, the council explains the harmful aspects in more detail:

"Science has shown that pesticide against unwanted insects also can entail that bees are dying. Neonicotinoids is a type of repellant that eliminates the central nervous system of insects and is used as pesticide against unwanted insects. These effective repellants are also believed to harm the central nervous system of bees and entail unnormal stimulus of the nerves, disorientation, and blocked receptors. The impacts on the nerves are incurable and affects the bee's behavior, how it is searching for food and how it navigates. This can lead up to the death of the bee." [38] (p. 24)

Furthermore, a relevant statement can be found that relates to the circumstantial analysis of the council:

"It should be noted that the repellants and pesticides that are in focus have significant shares on the market and that they probably also have had a big positive influence on the production of food globally" [38] (p. 24)

In terms of practical claims for action, the council points out that they "want to gain a deeper understanding of the scientific research and the level of scientific review as well as receiving guidance concerning what practical measures that the producers could employ" [38] (p. 24).

To make the council's reasoning explicit: The relevant repellants and pesticides are important in terms of constituting significant shares of the market, which in turn means that this market would suffer a significant blow if these products where taken off the market, which thereby could have negative effects on economies and "probably" in relation to food production in general. Thus, one set of business values stands against another set of business values, thereby neglecting the fact that bees and other insects or animals, that constitute vital parts of ecosystems, have an intrinsic value and enable various ecosystem services for both humans and animals. This business-oriented framing of the council is part of an economic discourse in which the market and the economy is the starting point for how problems such as bees being harmed by pesticides are to be managed.

To summarize, this analysis illustrates the nature of the council's practical reasoning, which is clearly pervaded by the discourse of economic rationalism. The council certainly addresses different values, including those associated with sustainable development, but this is largely done by translating them into the language of economic rationalism. The overarching goal of the council (and the fund system) is economic return, while other values are largely treated by the council as restrictions and risks for implementing the main goal. Thus, rather than explicitly assessing how and to what extent investments may lead to sustainable development, the focus is on how sustainability problems can increase financial risks that may negatively impact on business activities and economic return. In other words, the council is not committed to ethical deliberation in terms of balancing different values, like economic growth in relation to clean air. Instead, the dominant approach of the council and the funds is economic calculation in which sustainability problems are just one type of factor among others that are included in their calculations for economic return. However, this inclusion can anyhow be seen as an important step towards responsible investments.

## 5. Discussion and Conclusions

This article examined The Council on Ethics, which is the key unit of the Swedish public pension system for supporting the funds in seriously considering sustainable development in their investment activities. The formation of this council in 2007 by the AP Funds 1–4 should be seen against the background of more ambitious sustainability goals of the Swedish Government in governing the public pension system. The council and the four funds closely cooperate in developing more ethically responsible investments. This article examines the ethics of The Council on Ethics by empirically studying how its expression of words connected to public, environmental and business-related values have changed over time in its annual reports (2007, 2012, and 2018), as well as the practical reasoning of the

council in situations where value conflicts are present. Thus, the article is a longitudinal study that combines quantitative data with critical discourse analysis.

The results of the first research question—on how the council's expression of words have changed over time—show that words indicative of "financial business values" and "public procedural values" have decreased to a relatively large extent, while words associated with "sustainability values" have increased. Thus, over a period of more than ten years, the council has obviously adapted its language to the sustainability discourse. However, the implications of this can be discussed, and need to be approached in a more qualitative and critical sense. This is done in the second research question, focusing on the reasoning and argumentation of the council in the context of value conflicts. Two hypotheses guide this critical discourse analysis: the first one states that the council will make practical arguments for actions that explicitly consider and deliberate on sustainable development values, while the second assumes the council to argue in line with the discourse of economic rationalism, as developed by John Dryzek.

The results in relation to the second question show that the council often reasons in a rather general and loose way with respect to preferable solutions and possible ways forward, while more concrete claims for action are largely lacking. Furthermore, the article shows that the practical argumentation of relevance to sustainable development is somewhat contradictory in the sense that "top-down" compliance with international rules and principles are systematically argued for, while local experiences and solutions are highlighted as important at the same time. Most importantly, the overarching goal is economic return, while other values are largely treated by the council as restrictions and risks in relation to the main goal. Thus, it is about economic calculation rather than ethical deliberation, which should be seen against the background of the financial tradition of the funds and their overarching goal. The underlying discursive rationale of the council's practical reasoning is very much in line with the discourse of economic rationalism, which also persists over time. The findings point to a systematic use of the market as a basic entity that works as a starting point for the reasoning. The nature is assumed to be principally subordinated the economic system in terms of being valued primarily for its services and resources *for* the economy. Individuals are perceived as rational producers and consumers, motivated by material self-interest. To some extent, rhetorical devices in line with economic rationalism are used in the practical reasoning of the council.

The empirical results of the two first questions constitute the basis for answering the third one concerning eventual discrepancies between the quantitative and qualitative data. This comparison actually shows a substantial difference: the quantitative findings suggest an emerging sustainability discourse in which financial business values appear to be attributed less importance over time but, at the same time, the qualitative analysis shows that a continuous economic rationale underpins the practical argumentation of the council. Thus, both hypotheses presented in the theoretical section are supported, even though the second one regarding economic rationalism is more fundamentally confirmed, since the council's thinking and reasoning is pervaded by this discourse. Thus, the results of this article confirm previous studies, summarized in the introduction, when it comes to the increasing attention to sustainability among public pension funds in many countries, which thus seems to be a general phenomenon. Furthermore, this study contributes with a longitudinal design, and the results show that the discourse of sustainable development seems to be increasingly important over time, at least for the critical case of the Swedish pension system. However, thanks to the critical discourse analysis, this article also illuminates the discrepancies between manifest and latent levels of documents, underscoring the importance of using different methods in document studies. However, considering this limitation to document analysis, it is urgent that future studies also employ other methods and materials, such as interviews and observation studies, to get more insights into the internal processes of public pensions funds and their advisory units, like the Council on Ethics.

How can these results be interpreted? Is this a clear-cut case of greenwashing documents and letting "business as usual" persist? What might speak in favor of this interpretation is the increasing external pressure to adapt to the sustainability discourse, which is likely particularly challenging for organizations operating under a business culture of maximizing economic return. However, there is more to it. Even though economic return is the overarching goal of the public pension funds they do not act as venture capitalists. Public pension funds have rather a long-term perspective on economic return for the pension beneficiaries, which implies that a healthy environment and stable economic and political systems are of central concern. This long-term perspective on economic return further means a larger sensitivity to different types of risks, particularly those that can be very harmful in the longer time frame. However, the unpleasant side of this is that public pension funds, with very large amounts of economic assets, face major challenges in selling all of their unsustainable and unethical holdings on a short-term basis. Thus, this is the basic dilemma of pension funds in a world of unsustainable business activities. This also means that it is easy for critical journalists to find problematic business activities owned by pension funds, as was illustrated in the introduction of this article. This does not mean, however, that nothing happens. Instead of greenwashing, the metaphor of a "slow train coming" seems more appropriate, meaning that green institutional change takes time. This is also true for the Swedish case, even though it is perceived as a critical one in terms of being a frontrunner in the international process towards more responsible investments.

To increase the speed of green change surely requires additional external pressures from public policy makers and civic societies around the world to successively raise the risk premium on unsustainable investments. In such a process, market dynamics will of course play a key role as well. Citizen activities like green political consumerism, social activism, and voting can also increase the pressure on governments and public pensions funds to take sustainable development and responsible investment seriously. However, green institutional change and transformative learning may also follow from sudden disrupters and critical events, like pandemics, rapidly increasing temperatures, and dramatic turbulence on global markets [24,39]. However, let us hope that democratic mobilization and processes can do much of the needed work that we have in front of us.

**Author Contributions:** The quantitative and qualitative studies were accomplished by J.B. as well as the critical discourse analysis. The theoretical framing and writing of the article were done in cooperation between the two authors with equal contributions. All authors have read and agreed to the published version of the manuscript.

**Funding:** This research is part of the project *Green Public Ethics*, funded by the research council FORMAS in Sweden: https://formas.se/.

**Institutional Review Board Statement:** Not applicable.

**Informed Consent Statement:** Not applicable.

**Data Availability Statement:** Data available in a publicly accessible repository that does not issue DOIs. Publicly available datasets were analyzed in this study. This data can be found here: Etikrådet/Council on Ethics, https://etikradet.se/en/startsida-english/.

**Conflicts of Interest:** The authors declare no conflict of interest. The AP Funds and its Council on Ethics are Swedish public agencies, and the annual reports that are studied in this article are public documents under the Swedish principle of openness (Etikrådet, 2020): https://etikradet.se/en/startsida-english/.

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
