# Peer review of "Sustainability in Public Pension Funds? A Longitudinal Study of the Council on Ethics of the Swedish AP Funds"

_sustainability, doi:10.3390/su13010429_

Round 1
Reviewer 1 Report
It seems that authors in their text should explain to the potential reader the relation and influence of the Council of Ethics on pension funds and the role it plays or may play in the investment decisions of funds.
The overall conclusions of the study needs improvement. The authors referred to some of them superficially. It seems that the authors should also propose their own solutions. For example, one of the conclusion states: "The growing insight among public pension fund managers that unethical business activities and their consequences (child labor, weapon industry, fossil fuel) will likely come with a price that is difficult to estimate". It seems that apart from describing the full phenomenon and the behavior of the Council of Ethics, the authors should also propose their own solutions to the problem they noticed.
Author Response
Please, see the attachment

Reviewer 2 Report
In this paper the authors analyse if Swedish funds are considering seriously the issues linked to sustainability. The theme under analysis is interesting but I think the paper is not treated like a scientific article, failing in several issues, such as:
- I can start identifying that the paper does not follow the template of the journal. Font, section numbering, references, lack of cross-references are just some examples of the issues that authos should pay very attention in each submission.
- The way that the paper is organized do not help in the reading. For example, despite the fisrt section is the one called introduction, I believe tha the following pages also represente a introduction.
- The use of the methodologies should be correcly justified. For example, authors use Wordsmith but they have to describe better the tool and also identify other related or non-related works where the same method is applied.
- Over all the paper, the authors make several interrogations. With what purpose? From where came those questions? From author's experience? From earlier discussions? In this case, the way as the paper is organized and the way how these questions appear make the final result some confusing.
- Finally, I do not consider the conclusion as... conclusive. Mainly because the methodology is not well explained of because I cannot see the linkage between the results and the theme under analysis, I think this is also a critical issue of the paper.
Author Response
Please, see the attachment

Reviewer 3 Report
Referee report
Dear author(s):
In accordance with the revision of the article " Sustainability in Public Pension Funds? A longitudinal study of The Council on Ethics of the Swedish AP" (Manuscript ID: sustainability-1013209), which it has been entrusted to me,
Synopsis of the review
In general terms the author(s) has done a good job, using a fairly acceptable level of academic English, however, I consider that this work does not meet the minimum academic quality standard to be published in Sustainability:
- It could be said that until the section "Methods and materials", the work was going well, however, from here everything changes:
- - Line 270. Please use the US dollar, euro or any other currency that is usually used as a homogeneous reference for the following works. If you express the units in Swedish kroner, not all readers can get an idea of this amount, except those who are familiar with this particular currency.
- - I find no substantial differences between what the authors call "quantitative analysis" and "qualitative analysis". In lines 324-325 the authors used software packages. I think it is not so necessary to expand on the use of "Word" or "Wordsmith", since today there are many programs that do exactly the same thing.
- - The structure of the work is quite deficient. What is the point of placing the results of the alleged quantitative analysis in the appendices when they can be placed directly within the empirical part?
- - I do not consider the results to be sufficiently explanatory in figures 2, 3 and 4, nor do I consider it appropriate for the quantitative analysis to focus on obtaining them. Have the authors thought about the possibility of using a wordcloud?
- - The conclusions are not really "clear", taking into account that they are based on figures whose relevance does not say much to a future reader, nor are they based on a concrete scientific basis.
- - According to future works, have you thought about using bibliometrics or scientometrics techniques directly?
- - Finally, I advise against using ambiguous language. For example:
"The theoretical framing and writing of the article were done in cooperation between the two authors with approximately equal contributions". "Approximately" is not really conclusive.
Best regards,
The reviewer
Author Response
Please, see the attachment

Round 2
Reviewer 1 Report
ok
Author Response
Dear reviewer 1,
We are glad that you are satisfied with our revisions. Many thanks for constructive and very helpful comments!
The authors
Reviewer 2 Report
I consider that the authors did not make a serious review. They almost limited to change the name of the sections but in my opinion they answered very few of the comments of the reviewers. The conclusion is much better but I think that previous issues were not answered. I still have many doubts, for example, about the robustness of the methodology used in the paper.
Author Response
Dear reviewer 2,
Thanks for the comments, even though we expected more extended and well-motivated comments. We also miss interaction on our first reply. We raised questions because the review was very short and in parts difficult to understand. In doing major revisions you need to convince us with arguments what we should do and how it would help developing the article. This is not done so we are not prepared to make major revisions based on vague comments. Furthermore, we are surprised about the statement that we have not considered the other issues. We describe in detail in reply 1 how we have handled the previous comments. It is a long answer to a very short review.
If you as a reviewer “still have many doubts, for example, about the robustness of the methodology used in the paper”, please let us know what you mean. Are you questioning the quantitative or qualitative study or both? This questioning is in fact a new comment. In your first review you wanted us to better explain the method, which we have done in line with your comment by discussing wordsmith a bit more and adding references as well as motivating the use of relatively many citations to secure transparency in how we analyze the studied documents (see page 8). Transparency is quite important for qualitative analysis of practical reasoning. Is it something in how we do this qualitative analysis that you question or want to have improved? Or are you questioning the validity of our results? You have not properly communicated your queries, which makes it difficult to understand which type of revisions you really want and why. Still, we are of course grateful for some of comments that have helped us on the way.
Best wishes
The authors
Reviewer 3 Report
Referee report
Dear author(s):
In accordance with the revision of the article " Sustainability in Public Pension Funds? A longitudinal study of The Council on Ethics of the Swedish AP" (Manuscript ID: sustainability-1013209, version 2), which has been entrusted to me, I communicate below what has been my comments:
Synopsis of the review
More specifically:
Although the idea of the article is quite good, the definitive contribution of the authors to this field of knowledge, in my opinion, is somewhat limited. The article should be much more concise and "direct" for a future reader. I see that textual citations predominate from page 10 onwards, for example, in the lines:
* [405-411], [421-426], [436-446], [466-468], [488-494], [530-534], [545-547], [554-557], [577-583], [600-612], [614-621], [649-652], [654-660], [663-665].
Likewise, according to the authors themselves, figure 1 is the work of Fairclough and Fairclough [16], lines [136-151].
In conclusion, more than 10% of the article (considering the main body, lines 1-713) correspond to textual quotations in quotation marks or to this graphic.
By this, I do not mean under any circumstances that the author(s) have engaged in any kind of practice considered negligent; I simply mean that they should be considerably more concise with respect to the subject they are analyzing.
- Ambiguity in some citations
Line 115: According to Norman Fairclough, ...
Line 541: This can be seen in the light of Dryzek's component called assumptions ...
Line 712: discourse of economic rationalism, as developed by John Dryzek....
Etc
Regardless they were cited earlier or later, they should be properly referenced and included in the citation list.
- Material and Methods
I insist again on what I indicated in the previous review, it is not so important to extend the exposition in the computer application used, but the results obtained and, in fact, I insist again that to count in an analytical way the key terms today have multiple tools, not only "Wordsmith".
As for the results, figures 2-4, in my view, are not sufficiently explanatory for a future reader, hence I suggested the use of a word cloud, a universally known graphic representation to represent the key terms. Similarly, I consider the quantitative analysis to be incomplete: it is essential that author (s) had included a table relating to a summary the descriptive statistics.
- Discussion and Conclusion
I do not consider them purely a section of "Discussion and Conclusion" but a rather personal "essay", in which there is hardly any connection with the predominant literature. For example [lines 736-738 ]
"the results of this article confirm previous studies when it comes to the increasing attention to sustainability among public pension funds in many countries, which thus seems to be a general phenomenon".
What previous works? In any case, why don't you quote them?
Throughout this section, lines 691-773, if I have counted correctly, only two citations appear. I don't consider it proper, since this work would in principle could give for much more.
Honestly, I love the way writers express themselves, but a scientific article would have to be less passionate and more concise: Do not take this assessment as a lack of consideration at all.
I would also like to highlight that all the points I have mentioned have been done only with the aim of improving this manuscript.
Best regards,
The reviewer
Author Response
Dear reviewer 3,
Thanks for reviewing! Here comes our response.
Reviewer’s comments:
More specifically:
Although the idea of the article is quite good, the definitive contribution of the authors to this field of knowledge, in my opinion, is somewhat limited. The article should be much more concise and "direct" for a future reader. I see that textual citations predominate from page 10 onwards, for example, in the lines:
* [405-411], [421-426], [436-446], [466-468], [488-494], [530-534], [545-547], [554-557], [577-583], [600-612], [614-621], [649-652], [654-660], [663-665].
Likewise, according to the authors themselves, figure 1 is the work of Fairclough and Fairclough [16], lines [136-151].
In conclusion, more than 10% of the article (considering the main body, lines 1-713) correspond to textual quotations in quotation marks or to this graphic.
By this, I do not mean under any circumstances that the author(s) have engaged in any kind of practice considered negligent; I simply mean that they should be considerably more concise with respect to the subject they are analyzing.
Author response: Qualitative analysis of the type we do (critical discourse analysis) requires relatively long citations from the documents to secure transparency in how we analyze the texts. This is motivated in the method section (page 8). Does the reviewer disagree to this? If so, on what grounds?
Reviewer’s comments:
Ambiguity in some citations
Line 115: According to Norman Fairclough, ...
Line 541: This can be seen in the light of Dryzek's component called assumptions ...
Line 712: discourse of economic rationalism, as developed by John Dryzek....
Regardless they were cited earlier or later, they should be properly referenced and included in the citation list.
Author response: We can’t see any ambiguity in this. What does the reviewer mean by that? These are central references included in the reference list.
Reviewer’s comments:
Material and Methods
I insist again on what I indicated in the previous review, it is not so important to extend the exposition in the computer application used, but the results obtained and, in fact, I insist again that to count in an analytical way the key terms today have multiple tools, not only "Wordsmith".
As for the results, figures 2-4, in my view, are not sufficiently explanatory for a future reader, hence I suggested the use of a word cloud, a universally known graphic representation to represent the key terms. Similarly, I consider the quantitative analysis to be incomplete: it is essential that author (s) had included a table relating to a summary the descriptive statistics.
Author response: We have added comments and references in relation to wordsmith and of course there are other tools as well. Does the reviewer want us to mention alternative tools or….? We have already in our first response answered the question about figures 2-4 saying that the figures show valid results, and they are used mainly for illustration and should be seen in relation with the diagram 1. What is problematic with this answer? The statement about the need of a table summarizing descriptive statistics is a new comment from the reviewer. We do not see the point here. Is it in relation to the statistics in diagram 1 or?
Reviewer’s comments:
Discussion and Conclusion
I do not consider them purely a section of "Discussion and Conclusion" but a rather personal "essay", in which there is hardly any connection with the predominant literature. For example [lines 736-738 ]
"the results of this article confirm previous studies when it comes to the increasing attention to sustainability among public pension funds in many countries, which thus seems to be a general phenomenon".
What previous works? In any case, why don't you quote them?
Author response: We have summarized previous research in the introduction of the article, and we have added a short comment about that in the part that reviewer cites from the discussion/conclusion section so that the reader is informed about where previous research is discussed.
Throughout this section, lines 691-773, if I have counted correctly, only two citations appear. I don't consider it proper, since this work would in principle could give for much more.
Honestly, I love the way writers express themselves, but a scientific article would have to be less passionate and more concise: Do not take this assessment as a lack of consideration at all.
I would also like to highlight that all the points I have mentioned have been done only with the aim of improving this manuscript.
Author response: The comments above are difficult to respond to. Why should it be necessary with citations in the discussion/conclusion section? We do not see ourselves as passionate and the article is concise and well argued with valid results and clear conclusions. We appreciate of course that the reviewer “love” the way we express ourselves and we are grateful for the ambition to help us improve the manuscript.
Best wishes,
The authors